# Unusual Faces of Bladder Cancer

**DOI:** 10.3390/cancers12123706

**Published:** 2020-12-10

**Authors:** Claudia Manini, José I. López

**Affiliations:** 1Department of Pathology, San Giovanni Bosco Hospital, 10154 Turin, Italy; claudiamaninicm@gmail.com; 2Department of Pathology, Biocruces-Bizkaia Health Research Institute, Cruces University Hospital, Barakaldo, 48903 Bizkaia, Spain

**Keywords:** bladder cancer, diagnosis, differential diagnosis, prognosis, histopathology, immunohistochemistry

## Abstract

**Simple Summary:**

The spectrum of architectural and cytological findings in UC is wide, although transitional cell carcinoma, either papillary or flat, low- or high-grade, constitutes the majority of cases in routine practice. Some of these changes are just mere morphological variations, but others must be recognized since they have importance for the patient. The goal of this review is to compile this histological variability giving to the general pathologist a general idea of this morphological spectrum in a few pages. The review also updates the literature focusing specifically on the morphological and immunohistochemical clues useful for the diagnosis and some selected molecular studies with prognostic and/or diagnostic implications.

**Abstract:**

The overwhelming majority of bladder cancers are transitional cell carcinomas. Albeit mostly monotonous, carcinomas in the bladder may occasionally display a broad spectrum of histological features that should be recognized by pathologists because some of them represent a diagnostic problem and/or lead prognostic implications. Sometimes these features are focal in the context of conventional transitional cell carcinomas, but some others are generalized across the tumor making its recognition a challenge. For practical purposes, the review distributes the morphologic spectrum of changes in architecture and cytology. Thus, nested and large nested, micropapillary, myxoid stroma, small tubules and adenoma nephrogenic-like, microcystic, verrucous, and diffuse lymphoepithelioma-like, on one hand, and plasmacytoid, signet ring, basaloid-squamous, yolk-sac, trophoblastic, rhabdoid, lipid/lipoblastic, giant, clear, eosinophilic (oncocytoid), and sarcomatoid, on the other, are revisited. Key histological and immunohistochemical features useful in the differential diagnosis are mentioned. In selected cases, molecular data associated with the diagnosis, prognosis, and/or treatment are also included.

## 1. Introduction

Bladder cancer is a frequent neoplasm [1] in which tobacco use, pollution, and other varied agents have been directly implicated in its genesis and development [2]. Most of them are composed of transitional cells of low/intermediate grade, papillary architecture, and invasion limited to the lamina propria and submucosa. However, a smaller but significant number of cases do display dismal features like high-grade, non-papillary growth patterns, and muscularis propria invasion, with these patients pursuing an aggressive clinical course. 

Aside from transitional cell carcinoma (TCC), other histological subtypes, like conventional squamous cell carcinoma, adenocarcinoma, and neuroendocrine carcinoma, are quite frequently seen in clinical practice, alone or in combination, particularly in the context of high-grade cases. These cases are not the subject of this review.

Although TCC is a histologically monotonous neoplasm composed in the vast majority of cases by easily recognizable transitional cells, a small subset of cases displays a broad spectrum of architectural and/or cytological characteristics that should be recognized since some of them carry diagnostic difficulties and/or prognostic implications [3] (Table 1). This recognition is increasingly important now that very promising advances linking morphological variants with genomic signatures are being identified [4]. 

Clinical practice allows the pathologist to face unusual histological subtypes of urothelial carcinomas (UC), and conventional TCC displaying focal/extensive morphologic variations of uncertain significance. This narrative collects 25 years of personal experience of the authors in the routine diagnosis of bladder cancer. 

## 2. Architectural Changes

### 2.1. Nested and Large Nested Architecture

Talbert and Young reported in 1989 three cases of a deceptively benign bladder carcinoma characterized by small packed cellular aggregates closely resembling von Brunn nests and nephrogenic adenoma [5]. Isolated cases of this histological subtype of bladder cancer had previously appeared in the literature, always being referred to as of von Brunn nest origin [6]. Now, nested UC is well recognized and fully characterized by histological, immunohistochemical, and molecular perspectives [7,8]. Under the microscope, nested UC appears as a non-papillary neoplastic growth of bland cells with scarce atypia arranged in small nests (Figure 1a) showing an evident infiltrating growth pattern at different levels of the bladder wall. Typically, the tumor does not induce a stromal reaction nor is accompanied by inflammatory infiltrates. Problems to recognize nested UC may arise in small superficial biopsies if crushing artifacts are present or if the infiltrative nature is not seen. 

Cox and Epstein described in 2011 the large nested variant of UC reporting the characteristic histology of a tumor resembling large von Brunn nests with inverted growth in 23 patients [9]. Some isolated cases of this UC variant have been reported since then, and only two more series of cases have been published so far [10,11]. The large nested UC shares with the nested UC the same morphologic characteristics and clinical aggressiveness but the nests are larger (Figure 1b), with a growth pattern mimicking conventional inverted UC. These similarities have been advised to merge them into the same group in the last WHO classification of UC [12]. Interestingly, large nested UC displays a luminal phenotype, positive with FOXA1, GATA3, and CK 20 [12]. *FGFR3* and *TERT* genes are frequently mutated in this UC subtype [12].

### 2.2. Micropapillary Architecture

UC may sometimes display a micropapillary architecture. Delicate, thin, and fragile papillae without stromal axis are the hallmark of this morphological variant of UC (Figure 1c). To note, the invasive component of micropapillary UC shows nests with cells detached from the basal membrane, a typical artifact in this tumor that mimics lymphatic invasion and is associated with biological aggressiveness [13]. The vascular invasion is a very frequent histological finding (Figure 1d). Aside from rare pure examples, the majority of cases are mixed with a conventional transitional cell carcinoma, usually high grade. Like the rest of micropapillary carcinomas across the body [14], this histological subtype of UC has a dismal prognosis, even worse than conventional high-grade UC at the same stage [15], and typically presents with advanced stages at diagnosis. Although exceptional, micropapillary carcinomas from other sites may metastasize to the urinary bladder [16], making the correct diagnosis more difficult. 

The first description of this variant of UC was made in 1994 by Amin et al. [17], where they stressed the histological similarities of this bladder tumor with the classic papillary serous carcinoma of the ovary. After that, many series have been published all along the urinary tract, including the renal pelvis and ureter [18,19]. 

Abundant immunohistochemical and molecular analyses have been performed in micropapillary UC [20,21,22] all confirming its aggressive potential. Although initially thought to be a variant of adenocarcinoma by some authors [23], Yang et al. have very recently reported that the micropapillary UC is not a variant of adenocarcinoma [22]. 

### 2.3. Myxoid Stromal Change

UC may display focal myxoid changes in the stroma (Figure 1e) mimicking the colloid adenocarcinomas seen in other sites. This change has been previously reported [24,25] and when observed in transurethral resection specimens, may lead to an erroneous interpretation of colonic adenocarcinoma invading the bladder wall. Solid cell nests immersed in a basophilic edematous stroma are the hallmark of this histological change, which is usually focal but can be generalized in some isolated cases. Again, immunohistochemistry is of much help in cases in which the transitional phenotype of the tumor is not evident on hematoxylin-eosin slides. GATA3 positivity, co-expression of CK7 and CK20, and CDX-2 negativity should resolve the diagnostic dilemma in doubtful cases [24]. Attention must be paid, however, to the occasional CK7 positivity of some colorectal adenocarcinomas, a finding that is a sign of dismal prognosis [26].

### 2.4. Small Tubules and Nephrogenic Adenoma-Like Architecture

Very occasionally, UC is composed of low-grade cells arranged in small tubules resembling *cystitis glandularis* or nephrogenic adenoma (Figure 1f) [27]. The bland cytologic features of this histologic subtype contrast with its frank infiltrative nature, even reaching the muscularis propria in some cases. Since nephrogenic adenoma may display also a concerning pseudo-infiltrative growth [28], an immunohistochemical study with PAX-8, CK7, p63, and napsin A [29] may be useful to make the differential diagnosis in problematic cases. The clinical significance of this histologic change is not established so far. 

### 2.5. Microcystic Architecture

The microcystic histology has been rarely reported in the literature at UC. Aside from a handful of single case reports, the largest series published to date analyzes 20 cases [30]. The limited examples reported up to now show a bland histologic appearance, with round to oval cysts which often contain eosinophilic intraluminal secretion covered by low columnar or flattened urothelial cells (Figure 1g). Despite its deceptive bland histology, microcystic UC displays the same aggressiveness of conventional UC at the same stage. The main differential diagnosis is nephrogenic adenoma and adenocarcinoma of the bladder. In this sense, a basic immunohistochemical panel including p63 positivity and CK7/20 co-expression coupled with napsin A and PAX-8 negativities will resolve the eventual diagnostic troubles. 

### 2.6. Verrucous Architecture

Genuine verrucous carcinoma is a rare tumor subtype in the urinary tract [31], however, conventional well-differentiated squamous cell carcinoma with “verrucous” architectural features is a much more common event. Since the difference between them has prognostic implications their correct identification by the pathologist matters. Verrucous carcinoma may recur but never metastasize. Some cases are related to HPV infection, others to schistosomiasis, but there are also cases unrelated to any known specific etiology [32].

The diagnosis of a verrucous carcinoma in the urinary tract, as elsewhere, is subjected to very strict histological criteria. Only low-grade keratinizing squamous cell carcinomas with superficial verrucous architecture should be considered (Figure 1h). Verrucous carcinomas may display a pushing border of growth into the lamina propria, but a true invasion is lacking. Noteworthy, any high-grade area across the tumor or frank stromal infiltration makes the diagnosis of verrucous carcinoma unsuitable. 

### 2.7. Diffuse Architecture with Lymphoepithelioma-Like Changes

Lymphoepithelioma is the classical histological term referring to an undifferentiated carcinoma first described in the nasopharyngeal region of Asian patients [33]. Some of them are related to Epstein-Barr virus infection. Since then, analog histology has been described in many carcinomas widely distributed in the body. Aside from multiple case reports, several series of this tumor subtype in the bladder [33,34,35,36] and the upper urinary tract [37] have been published in the literature. Remarkably, the theoretical relationship of lymphoepithelioma-like UC with Epstein–Barr virus infection is no longer sustainable in cases arising in the urinary tract after the results obtained with FISH analyses in the largest series [35,36,37].

The tumor shows a diffuse growth of ill-defined islands of poorly-differentiated cells with badly defined cytoplasmic borders, large nuclei, and patent nucleoli. The stroma is heavily infiltrated by lymphocytes occasionally showing lymphoepithelial lesion (Figure 1i). By immunohistochemistry, GATA3, cytokeratins 34βE12, AE1-AE3, and CK7, p53 and p63 are positive in a variable number of cases, whereas TTF-1, CD30, and CK20 are negative [36,37]. The prognosis does not differ from conventional UC at the same stage.

## 3. Cytological Changes

### 3.1. Plasmacytoid Cells

Plasmacytoid UC is an aggressive tumor. This cytologic variant of UC can present as pure tumors or mixed with conventional UC and/or with other non-conventional UC. For example, mixed micropapillary and plasmacytoid UC cases have been occasionally reported [38]. Histological similarities with multiple myeloma were noticed since the first report by Sahin et al. in 1991 [39]. Since this original description, several large series have been published so far all of them confirming its dismal prognosis [40].

In its typical presentation, the tumor appears as flat, non-papillary, highly cellular masses growing diffusely in the urinary tract wall with infiltrative edges and frequent vascular invasion images. Neoplastic cells are non-cohesively arranged and show lateralized cytoplasm, nuclear atypia, and high mitotic count (Figure 2a). In doubtful cases, or patients with a previous history of plasma cell dyscrasia, immunohistochemistry is of help revealing its epithelial, non-plasmacytic, nature. Briefly, GATA-3 and CK7 are positive and CD 38 is negative. Positive immunostaining with CD 138 may be observed in this neoplasm, but this finding does not preclude the diagnosis of plasmacytoid UC [41]. 

*HER2* overexpression has been observed by FISH in plasmacytoid UC [42]. Contrary to what happens in most UC, plasmacytoid variants do not seem to harbor *TP53* gene mutations in a sequencing analysis [41]. On the other hand, *TERT* gene promoter mutations have been detected [43]. A study using whole-exome sequencing has detected somatic alterations in the *CDH1* gene of 84% of plasmacytoid UC, a finding of clinical aggressiveness that seems to be specific to this tumor variant [44]. 

### 3.2. Signet-Ring Cells

Since signet-ring cell features are very rare in UC, and their identification in transurethral resection specimens can raise the possibility of a metastatic seed from a neoplasm originating in the digestive tract. A careful search of the classical urothelial features (nests of transitional cells, papillae, in situ carcinoma in the surface epithelium, etc.) in the biopsy, if present, may be of help in the differential diagnosis. Otherwise, the clinical context of the patient and a basic immunohistochemical panel, for example, CK7/20, GATA-3, CDX-2, and p63, should resolve the dilemma. The analysis of the national Surveillance, Epidemiology, and End Results (SEER) database of 318 such cases confirms the worse prognosis of this histologic variant compared with conventional UC [45].

### 3.3. Basaloid-Squamous Cells

Basaloid-squamous cell carcinomas are aggressive neoplasms mainly located in the head and neck [46] and anal [47] regions. The tumor is extraordinarily uncommon in the urinary tract, with only a handful of single cases published to date [48,49,50,51]. Everywhere, most basaloid squamous cell carcinomas are associated with HPV infection [51]. 

Histologically, the tumor is deeply infiltrative and shows a typical biphasic pattern (Figure 2b). Basaloid atypical cells with high mitotic rate and scarce cytoplasm are arranged in lobes and nests showing peripheral palisading and stromal reaction. Basaloid nests are centered by squamous islands with evident keratinization. p16 is intensely positive in tumor cells.

### 3.4. Yolk Sac Cells

A very limited number of UC with yolk sac tumor differentiation has been reported in the literature [52,53,54,55]. The yolk sac differentiation represents an example of a somatic differentiation present in non-gonadal neoplasms [56]. A varied spectrum of patterns have been identified in these tumors: microcystic, vitelline, glandular enteric-like, hepatoid, solid, sarcomatoid, etc. An enteroblastic differentiation seems to be the most frequent histology in somatically derived yolk sac tumors [56]. 

Immunohistochemistry is useful to identify yolk sac differentiation in UC and other somatic tumors considering the wide spectrum of patterns that can be detected in this tumor. Alpha-fetoprotein and SALL4 are consistently positive. CK7, however, is negative. Markers of intestinal differentiation, like CDX2, are usually positive in enteroblastic areas and Her Par-1 in hepatoid ones. A polysomic abnormality in 12p has been detected in one recently published case [55]. 

### 3.5. Trophoblastic Cells

Trophoblastic differentiation is a rare event in UC that has been recently reviewed by Przybycin et al. in a series of 16 cases [57]. The spectrum includes isolated syncytiotrophoblast cells interspersed in a conventional UC, mixed choriocarcinoma and UC, and pure choriocarcinoma. Same as in the yolk sac differentiation, trophoblastic changes are examples of somatically derived differentiations in non-gonadal tumors.

Syncytiotrophoblasts are detected as isolated multinucleated giant cells immersed in high-grade UC (Figure 2c). Choriocarcinoma differentiation appears as hemorrhagic areas at low-power magnification. A closer view of these areas reveals the typical mixture of trophoblastic and syncytiotrophoblastic cells immersed in a necro-hemorrhagic background (Figure 2d). 

By immunohistochemistry, β-hCG is expressed in trophoblastic and syncytiotrophoblastic cells, as well as in the malignant urothelial cells in a significant number of cases. Interestingly, increased levels of seric β-hCG in patients with UC is an independent prognostic factor [58]. GATA3 positivity has been detected in more than 70% of trophoblastic tumors in a large series [59] and appears as a useful marker to be included in the diagnostic panel. SALL4 is focally positive in less than 50% of the cases [57] and is negative in the larger syncytiotrophoblastic cells [60]. HSD3B1, a novel marker specific to trophoblastic differentiation [61], has been detected in 100% of the cases [57]. 

### 3.6. Rhabdoid Cells

Rhabdoid tumors have been documented in many different topographies across the body [62], always linked to biological aggressiveness and bad prognosis. Its histogenesis is still unclear. A handful of rhabdoid tumors of the bladder have been published, particularly in children and young adults [63,64,65]. There are, however, isolated cases reported in adulthood [66,67,68].

Aside from genuine rhabdoid tumors, a focal *rhabdoid* phenotype can be observed sometimes in UC [68], where large and ovoid cells with large atypical nuclei and lateralized eosinophilic cytoplasm may appear growing without any specific pattern usually in high-grade neoplasms. A possible rhabdomyoblastic dedifferentiation in the context of a sarcomatoid UC should be ruled out, at least theoretically, in these cases. 

By immunohistochemistry, rhabdoid cells are positive for CK7, CK20, vimentin, E-cadherin, and β-catenin, p63, and INI-1 [68].

### 3.7. Lipid/Lipoblast-Like Cells

These two terms refer to a rare variant of UC composed of lipidic appearing tumor cells intermingled with transitional cells in variable proportions. It was first recognized by Mostofi et al. in 1999 [69]. Since then only single case reports and two short series [70,71] have been published. The longest series so far analyzes 27 cases collected from different international institutions [71]. Lipidic-appearing cells may resemble either adipocytes or adipoblasts (Figure 2e) and usually take part in a high-grade UC, not otherwise specified. Immunohistochemistry confirms the epithelial nature in all cases, including the co-expression of CK7 and CK20 [70,71].

### 3.8. Giant Cells

Giant cells are rarely observed in UC and only single case reports and a few short series have been published so far [72,73,74]. Two morphological variants have been described: osteoclast-like and giant pleomorphic cells, both of them associated with high-grade neoplasms. For practical purposes, these cells must be distinguished from trophoblastic and syncytiotrophoblastic cells appearing in some UC (see above). The presence of these giant cells in UC may be focal in the context of a high-grade UC or diffuse across the tumor, making difficult the correct diagnosis. A dedifferentiated sarcomatoid UC (see below) diagnosis can be considered in some of these cases.

Pleomorphic giant cell carcinoma have been described in many sites of the body and is a tumor subtype with dismal prognosis everywhere. Giant cell tumor areas in UC show a diffuse growth of cells with extreme pleomorphism and high mitotic count (Figure 2f). Cytoplasmic vacuolization and emperipolesis can be detected. Unusually, these tumors are at advanced stages at diagnosis, with deep infiltration in the bladder wall and frequent lymphatic dissemination [72]. Fifty percent of the patients reported in the series of Samaratunga et al. died of disease within the first year of follow-up [73]. By immunohistochemistry, the co-expression of CK7/20 and GATA3 positivity are retained in these tumors. 

Osteoclast-like giant cells can be rarely observed in tumors originating in many sites of the body. In the bladder, they appear very occasionally in the context of high-grade UC. Contrary to the observed in pleomorphic giant cells, osteoclastic-like giant cells devoid of atypia and mitosis and show a reactive, non-neoplastic appearance. Whether these cells are truly neoplastic or reactive in the context of the tumor is a classical controversy that has been recently elucidated [74]. In this study, osteoclast-like giant cells are negative for GATA3, thrombomodulin, uroplakin II, and cytokeratin AE1/AE3, thus confirming their non-epithelial differentiation [74]. 

### 3.9. Clear Cells

Only single cases and a short series of 10 cases [75] of clear cell UC have been published so far. An advanced stage at diagnosis and an aggressive clinical course is the rule in these patients. Clear cell change, however, is regularly mentioned in several papers reviewing the varied morphology of UC in the bladder and upper urinary tract [76,77,78,79,80]. 

Clear cell change in UC reflects intracytoplasmic glycogen accumulation that in some cases is extreme this way resembling the typical clear cells observed in clear cell renal cell carcinoma. Usually, clear cell nests are intermingled in the tumor with conventional transitional cells (Figure 2g), which makes its correct identification easier. However, if the clear cell change is generalized or if transurethral resection specimens do not contain pieces of evidence of the urothelial origin of the tumor, the possibility of metastasis in the bladder of a clear cell renal cell carcinoma should be always taken into account [81].

### 3.10. Eosinophilic (Oncocytoid) Cells

An eosinophilic change can be observed in some UC resembling the cells of renal oncocytomas [82]. These cases show large granular and deeply eosinophilic elements with focal apocrinoid features (Figure 2h). Frequent nuclear pleomorphism is also seen, but true atypia is lacking. Mitoses are scarce, or absent, giving an overall impression of a low-grade tumor. Immunohistochemistry is that of the conventional UC, and neuroendocrine markers are negative. Anyway, further descriptions are needed to delineate better this histologic feature. At least for practical purposes, this histologic feature should be distinguished from oncocytic carcinoid tumors of the urinary bladder [83], an extraordinarily rare entity in the urinary bladder.

### 3.11. Sarcomatoid Cells

Sarcomatoid dedifferentiation is a relatively common finding in high-grade UC. Recent studies have approached the correlation between morphology and genomics in sarcomatoid bladder cancer through analyzing the epithelial to mesenchymal transition process concluding that UC developing sarcomatoid transformation are carcinomas of basal-type [84]. Practically all possible differentiations have been reported in the literature, from undifferentiated spindle cell (Figure 2i) to osteosarcoma. The epithelial component may be scarce or even not identified in some cases, so the diagnosis of primary sarcoma in the urinary bladder should be made with caution in transurethral resection specimens. 

An excellent review of this topic based on a MEDLINE database study has been recently published [85]. 

## 4. Conclusions

This narrative collects the varied spectrum of morphological features that can be found in UC. These changes have been organized in architectural and cytological for didactic purposes, but mixtures of them are eventually found in real practice. The goal of this overview is to offer in a few pages the essentials for recognizing them giving diagnostic clues based on morphological and immunohistochemical keys. 

## Figures and Tables

**Figure 1 cancers-12-03706-f001:**
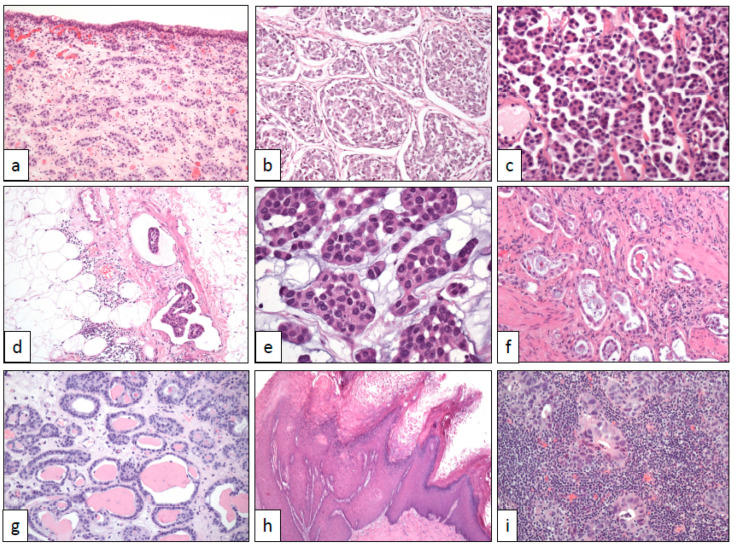
Architectural changes in bladder cancer (with original magnifications included). (**a**) Nested pattern (×100), (**b**) large nested pattern (×100), (**c**) micropapillary pattern (×250), (**d**) vascular invasion in the micropapillary pattern (×100), (**e**) myxoid basophilic stroma (×400), (**f**) small tubules (×100), (**g**) microcystic pattern (×100), (**h**) verrucous pattern (×40), and (**i**) lymphoepithelioma-like pattern (×250).

**Figure 2 cancers-12-03706-f002:**
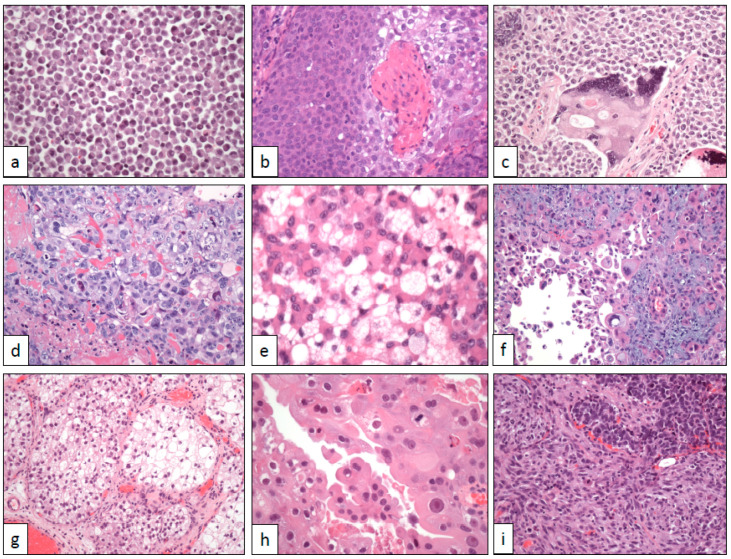
Cytological changes in bladder cancer. (**a**) Plasmacytoid cells (×250), (**b**) basaloid and squamous cells (×250), (**c**) syncytiotrophoblastic cells (×250), (**d**) trophoblastic cells (×250), (**e**) lipoblastic-like cells (×400), (**f**) pleomorphic giant cells (×250), (**g**) clear cells (×100), (**h**) eosinophilic (oncocytoid) cells (×400), and (**i**) sarcomatoid cells (×250).

**Table 1 cancers-12-03706-t001:** Unusual features in bladder cancer with prognostic profiles.

**Architectural Changes**	**Prognostic Profiles**
	**Worse prognosis** - Nested - Large nested - Micropapillary
	**Not worse prognosis** - Myxoid stromal change - Small tubules - Nephrogenic adenoma-like - Microcystic - Verrucous - Diffuse lymphoepithelioma-like
Cytological changes	
	**Worse prognosis** - Plasmacytoid - Signet-ring - Basaloid-squamous - Yolk-sac -Trophoblastic - Rhabdoid - Giant pleomorphic - Clear - Sarcomatoid
	**Not worse prognosis** - Lipid/lipoblast - Giant osteoclast-like - Eosinophilic (oncocytoid)

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
