# Peer review of "Unusual Faces of Bladder Cancer"

_cancers, 2020, doi:10.3390/cancers12123706_

Round 1

Reviewer 1 Report

  1. This manuscript is interesting and well-done.
  2. Theme of this research, to suggest a guideline between morphological and immunohistochemical clues useful for the diagnosis, and on some selected molecular studies with prognostic and/or diagnostic implications on bladder cancer. However, all authors should be cautious about “results were easy to anticipate”.
  3. Nevertheless, this manuscript was well-organized and interesting. So only minor points to be considered: English language should be revised throughout the text.
  4. 2p, line 52:`This narrative collects 25 years…’ is there any papers to reference?
  5. 5p, line 153~154: It's just my opinion.. If there is a known paper, these sentences need references, too.
  6. This paper is well organized for readers to understand. A main theme is so interesting too.

Author Response

Thank you for your evaluation

English has been improved throughout the entire manuscript

2p, line 52. We have decided not to include our references here to avoid unnecessary self-citations

5p, line 153-154. We have added there the reference #33.

Reviewer 2 Report

The current narrative review aims to report architectural and cytological characteristics about rare bladder cancer variants.

Authors should be commended for their intent. The work is well structured and gives to the reader a good perspective of uncommon pathologies, usually leave behind because their rarity.

Major comments:

1) Despite the work provides a detailed overview for an audience of pathologists, it sounds clear enough also for urologists interested not only in scratching the surface of urothelial bladder cancer, but also to have a deeper and wider knowledge of bladder cancers.

Nevertheless, it could be of great use for both pathologist and urologist to produce a table where architectural and cytological features are reported for each variant.

2) Moreover, it could be very interesting to report, whenever possible, in the text and/or in the table the aggressiveness/prognosis of each variant (just a simple discrimination between low/intermediate/high grade of aggressiveness or poor/intermediate/good prognosis).

Summarizing the two comments, a good table could help readers to have a faster and easiest access to the data after a first read and help the paper to be taken in consideration for citation in the future.

Author Response

Thank you for your evaluation

English has been improved throughout the entire manuscript.

A table summarizing the varied unusual faces of bladder cancer, with their respective prognosis, has been included.

Round 2

Reviewer 2 Report

Authors positively replied to my comments.

No more comments from my side.